# Efficient Hydrolysis of Gluten-Derived Celiac Disease-Triggering Immunogenic Peptides by a Bacterial Serine Protease from *Burkholderia gladioli*

**DOI:** 10.3390/biom11030451

**Published:** 2021-03-17

**Authors:** Yu-You Liu, Cheng-Cheng Lee, Jun-Hao Hsu, Wei-Ming Leu, Menghsiao Meng

**Affiliations:** 1Program in Microbial Genomics, National Chung Hsing University and Academia Sinica, Taichung 40227, Taiwan; jojo89115@gmail.com; 2Graduate Institute of Biotechnology, National Chung Hsing University, 250 Kuo-Kuang Rd., Taichung 40227, Taiwan; lichanjan@gmail.com (C.-C.L.); adg04789@gmail.com (J.-H.H.); wmleu@nchu.edu.tw (W.-M.L.)

**Keywords:** celiac disease, gluten-free diet, gluten intolerance, gluten-digesting endopeptidase, *Burkholderia gladioli*

## Abstract

Celiac disease is an autoimmune disorder triggered by toxic peptides derived from incompletely digested glutens in the stomach. Peptidases that can digest the toxic peptides may formulate an oral enzyme therapy to improve the patients’ health condition. Bga1903 is a serine endopeptidase secreted by *Burkholderia gladioli*. The preproprotein of Bga1903 consists of an N-terminal signal peptide, a propeptide region, and an enzymatic domain that belongs to the S8 subfamily. Bga1903 could be secreted into the culture medium when it was expressed in *E. coli.* The purified Bga1903 is capable of hydrolyzing the gluten-derived toxic peptides, such as the 33- and 26-mer peptides, with the preference for the peptide bonds at the carbonyl site of glutamine (P1 position). The kinetic assay of Bga1903 toward the chromogenic substrate Z-HPQ-*p*NA at 37 °C, pH 7.0, suggests that the values of *K*m and *k_cat_* are 0.44 ± 0.1 mM and 17.8 ± 0.4 s^−1^, respectively. The addition of Bga1903 in the wort during the fermentation step of beer could help in making gluten-free beer. In summary, Bga1903 is usable to reduce the gluten content in processed foods and represents a good candidate for protein engineering/modification aimed to efficiently digest the gluten at the gastric condition.

## 1. Introduction

Gluten refers to the water-insoluble proteins in the flour of wheat, barley, and rye, and it may account for 70%–80% of the flour proteins [1]. Among the gluten proteins, the ethanol soluble ones are called prolamins because they have high contents of proline and glutamine. The prolamins in wheat, barley, and rye are specifically called gliadin, hordein, and secalin, respectively. In general, gluten in the flour is a valuable nutrient. However, it may cause health problems to certain people. According to clinical symptoms and pathological mechanisms, the discomforts and diseases caused by gluten consumption can be classified into celiac disease, wheat allergy, and non-celiac gluten sensitivity (NCGS) [2,3,4].

Celiac disease is an inherited autoimmune disease triggered by the ingestion of gluten, and it develops only in individuals who carry either HLA-DQ2 and/or HLA-DQ8 allele [5,6]. Anti-transglutaminase 2 IgA and anti-deamidated gliadin IgG in serum can be used as the diagnostic biomarkers of the disease [7,8]. The major pathological features of celiac disease are intestinal villous atrophy and crypt hyperplasia [9]. Celiac disease is multifaceted because it has typical gastrointestinal and extraintestinal manifestations [10]. The former ones include abdominal cramping, stomach bloating, vomiting and chronic diarrhea, and the latter ones include anemia, osteoporosis, dental enamel hypoplasia, rash, infertility, arthritis and seizure. The prevalence of celiac disease in Caucasian is about 1%, which is 4–8 times higher in comparison with other races in the US [11,12]. Currently, the only treatment for celiac disease patients is a lifelong strict gluten-free diet; nonetheless, it is not always effective and places a heavy social and economic burden. Wheat allergy, by contrast, is an IgE-mediated acute hypersensitivity response to gluten [3]. The prevalence of wheat allergy is approximately 1% of the world population and it occurs mainly in children [13]. NCGS describes a gluten ingestion-related disease with varied symptoms, but it is neither celiac disease nor wheat allergy. Due to the lack of diagnostic biomarkers, it is difficult to estimate accurately the prevalence of NCGS [4].

Prolamins cannot be fully hydrolyzed by pepsin in the stomach because of the repetitive proline residues. Certain incompletely digested peptides would be presented by HLA-DQ2/HLA-DQ8-carrying dendritic cells after the glutamine residues are deaminated by tissue transglutaminase 2 in the lamina propria of duodenum [14]. This would trigger pro-inflammatory T cell responses, and lead to the production of autoantibodies against tissue transglutaminase 2. The prominent and well-known celiac disease-immunogenic peptides include the 33- and 26-mer peptides derived from α2- and γ5-gliadins, respectively. An oral enzyme therapy using gluten-specific peptidases after meals may provide an alternative to ameliorate the health condition of people who suffer from celiac disease as well as other gluten-related disorders. 

A peptidase exhibiting activity for gliadin hydrolysis at acidic condition was identified in the culture medium of a *Burkholderia gladioli* strain in this study. Subsequently, the corresponding gene was cloned, and the recombinant peptidase was expressed in *E. coli* BL21(DE3). The catalytic activity and substrate preference of the purified *E. coli*-expressed peptidase were characterized. The results indicate that this peptidase is able to digest the immunogenic peptides with a preference for peptide bonds after glutamine residue. The usefulness of this recombinant peptidase in making gluten-free beer was also demonstrated in this study.

## 2. Materials and Methods

### 2.1. Chemicals and Reagents

Gliadins and bovine serum albumin (BSA) were purchased from Sigma-Aldrich (St. Louis, MO, USA). Gliadin-derived 33-mer and 26-mer peptides were chemically synthesized by Mission Biotech (Taipei, Taiwan). Chromogenic peptidyl substrates, attached with a benzyloxycarbonyl group (Z) and a *p*-nitroaniline group (*p*NA) at the N and C termini, respectively, were synthesized by Kelowna International Scientific (Taipei, Taiwan). The product of acid peptidases isolated from the culture of *Aspergillus niger* was purchased from Taobao, China.

### 2.2. Bacterial Strains and Media

Bacterial strains able to secrete gliadin-hydrolyzing peptidases were isolated using gliadin-containing minimal medium plates (6 g/L NaH_2_PO_4_·H_2_O [pH 5], 1.5 g/L NaCl, 0.2 g/L KCl, 1 g/L MgSO_4_·7H_2_O, 0.2 g/L yeast extract, 0.1 g/L SDS, 0.9 g/L gliadin, and 15 g/L agar). The colony exhibiting a clear surrounding zone on the agar plate was considered a positive strain. To obtain bacterial broth with proteolytic activity, the selected strains were cultivated in the same minimal medium except gliadin was replaced with 20 g/L skim milk and agar was omitted. The cultivation was performed aerobically at 28 °C for 2 days. 

*E. coli* BL21(DE3) was used as the host to produce the recombinant peptidase. Lysogeny broth (LB) was used as the medium for routine culture of *E. coli*, while LM broth (5 g/L tryptone, 2.5 g/L yeast extract, 4.78 g/L Na_2_HPO_4_, 2.99 g/L KH_2_PO_4_, 5.5g/L NaCl, 0.05 g/L NH_4_Cl, 4 g/L glucose, 0.12 g/L MgSO_4_·7H_2_O, 0.033 g/L CaCl_2_·2H_2_O, 1 µg/L biotin, and 1 µg/L thiamin) was used to produce the recombinant peptidase.

### 2.3. Plasmids

The gene encoding this bacterial peptidase, termed *Bga1903*, was amplified by PCR from the chromosome of the *B. gladioli* strain using a pair of primers [5′-TTGATCCATGGATCAATTAGTTCGCACTACTTCT-3′ and 5′-TAAGGATCCTTACTGACGTGCCGCGTTGA-3′]. The nucleotides underlined are engineered restriction sites of *Nco*I and *Bam*HI, by which the amplified *Bga1903* was inserted into plasmid pETDuet-1 (Novagen, Madison, WI, USA). The peptidase expression in *E. coli* was poor based on this expression plasmid. Therefore, a codon-optimized version of the *Bga1903* gene (named *Bga1903+*) was chemically synthesized by Genewiz Inc. (Suzhou, China) according to the codon preferences of *E. coli*. The resulting plasmid pETDuet-*Bga1903+* was also engineered to have a hexahistidine-coding sequence attached to the 3′ end of *Bga1903+*. *E. coli* BL21(DE3) carrying pETDuet-*Bga1903+* was then used to produce the recombinant peptidase.

### 2.4. Protein Expression and Purification

*E. coli* BL21(DE3) harboring pETDuet-*Bga1903+* was grown aerobically in LB, supplemented with 100 μg/mL ampicillin, at 37 °C to an OD_600_ ≈ 0.8. At the time, IPTG was added into the culture to a final concentration of 1 mM and the cultivation was continued at 28 °C for 18 h. The culture was centrifuged at 10,000× *g* for 15 min at 4 °C, and the cell-free supernatant was harvested. This protein solution was 100-fold concentrated by using the Millipore Labscale TFF system, equipped with Pellicon XL cassette (Biomax 5 kDa). The concentrated solution was loaded into a HisTrap excel 5 mL column (GE Healthcare Life Science, Marlborough, MA, USA), followed by an intensive wash with PBS buffer (20 mM NaH_2_PO_4_, 0.5 M NaCl, pH 7.4). Finally, the protein bound on the resin was eluted with 100 mM imidazole-containing PBS buffer. The concentration of the purified protein was measured using the Bradford protein assay reagent (Thermo Fisher Scientific, Waltham, MA, USA) with BSA as the standard.

### 2.5. Zymogram Assay

The gliadin zymogram assay was performed as the standard SDS-polyacrylamide gel electrophoresis (PAGE) but with a couple of modifications, including (1) the 12% polyacrylamide gel contained 2.2 mg/mL gliadin, (2) the protein sample was loaded into the gel without prior heating at 95 °C, and (3) the gel after electrophoresis was incubated twice in renature buffer (100 mM Tris-HCl (pH 5.0) and 2.5% (*v*/*v*) Triton X-100) at 4 °C for 30 min, and subsequently in reaction buffer (100 mM Tris-HCl [pH 5.0] and 1% (*v*/*v*) Triton X-100) at 37 °C for 1 h. Finally, the gel was stained with Coomassie Brilliant Blue R-250 as in the regular procedure.

### 2.6. Enzymatic Activity Assay

The hydrolysis of gliadin was carried out by incubating gliadin (7.5 mg/mL) with the purified peptidase (0.25 mg/mL) in 25 mM glycine-HCl buffer at pH 2.5–3.5 for the indicated periods. The degradation pattern of gliadin was then analyzed by SDS-PAGE.

The removal of epitopes from gliadin was carried out in 0.1 mL solution that contained 60 μg gliadin, 1.2 μg of the purified peptidase, and 50 mM citrate-phosphate buffer at pH 3.0–7.0 at 37 °C for 1 h. The removal of gliadin in each of the reaction was assayed by the competitive ELISA kit using R5 monoclonal antibodies according to the instruction of the manufacturer (R-Biopharm AG, Darmstadt, Germany).

The hydrolysis of the 33- and 26-mer immunogenic peptides was carried out by incubating the peptide (1 mg/mL) and the purified peptidase (50 μg/mL) in 50 mM citrate-phosphate buffer (pH 6.0) at 37 °C for 3 h, followed by incubation at 95 °C for 5 min. The degradation degree of the peptide was analyzed with a Gilson HPLC system (Middleton, WI, USA) using a C18 column (Ascentis Express 25 cm × 4.6 mm, 5 μm, Supelco, Bellefonte, PA, USA). After sample injection, the C18 column was subjected to H_2_O with 0.1% (*v*/*v*) trifluoroacetic acid at a flow rate of 1 mL/min for 5 min, and then proceeded with linear gradient elution mode from 0%–80% acetonitrile with 0.1% (*v*/*v*) trifluoroacetic acid at a flow rate of 1 mL/min for 20 min. The cleaved fragments of the immunogenic peptides were subjected to LC-tandem mass spectrometric analysis (Triple TOF 6600 & QSTAR Elite, Applied Biosystems Sciex, Framingham, MA, USA). The results were viewed with Mascot software (Matrix Science, London, UK) with important parameter settings as follows: mass values, monoisotopic; peptide mass tolerance, ±0.05 Da; fragment mass tolerance, ±0.03 Da; and maximal missed cleavages, 0. To determine the peptide bonds on the 33- and 26-mer peptides that were cleaved by the purified peptidase, spectral data was searched against amino acid sequences of the peptides.

The enzymatic activity toward various chromogenic peptidyl substrates was measured according to the release of *p*-nitroaniline from the substrates. One activity unit was defined as the release of 1 nmol *p*-nitroaniline per second at the indicated conditions. Unless otherwise specified, the reaction containing 0.5 mM peptidyl substrates, 12.5 μg/mL of the purified enzyme, 1.5% (*v*/*v*) Tween 20, and 40 mM citrate-phosphate buffer (pH 7.0) was performed at 37 °C. The catalytic rate was calculated following the time-course increment of optical density at 405 nm.

### 2.7. Determination of Substrate Preferences of Bga1903

The preferential residues at the P1, P2, and P3 positions were determined by using BSA as the catalytic substrate. The hydrolysis of BSA was performed by incubating BSA (0.8 mg/mL) with the purified Bga1903 (80 μg/mL) in 50 mM citrate-phosphate buffer [pH 6.0] at 37 °C for 1 h. The small degraded fragments of BSA were collected by using a filtration device with a cutoff value of 10 kDa, and subsequently subjected to LC-tandem mass spectrometric analysis under the parameter settings as following: mass values, monoisotopic; peptide mass tolerance, ±2 Da; fragment mass tolerance, ±0.5 Da; and maximal missed cleavages, 0. Spectral data was searched against the amino acid sequence of BSA, revealing the compositions of detected proteolytic fragments. Every amino acid residue at the P1 position (the last residue of the fragment) was counted, and the frequency was expressed as the percentage of the total counts of all P1 residues. The probability of a given residue at the P1 position was then obtained after normalization of the frequency number by its abundance in BSA. The probability of a given amino acid at the P2 or P3 was obtained following the same calculation principle.

### 2.8. Beer Brewing

Barley grain (225 g) and wheat malt (50 g) were mixed into 1.2 L warm water (43 °C), followed by sequential heating at 68 °C for 1 h, 76 °C for 10 min and 100 °C for 15 min. Then, 1.4 g hop was added and the extracted mixture (wort) was boiled continuously for 1 h. The wort was cooled down to room temperature, followed by inoculation of 7 g dry wheat beer yeast (Safbrew WB-06, Fermentis, Marcq-en-Baroeul, France). The fermentation was kept at 20 °C for 7 days. Finally, the broth was clarified by centrifugation at 2500× *g*. To see the reducing effect on the gluten content in beer, acid peptidases from *A. niger* and the purified Bga1903 at the indicated amounts were included in the wort at the time when the yeast was added.

## 3. Results

### 3.1. Identification of Bacterial Gliadin-Hydrolyzing Peptidases

To find peptidases with the potential to treat celiac disease, microbial specimens from a variety of habitats, including soils, plant materials, and insect gut, were spread on gliadin-containing minimal medium agar plates, pH 5.0, as described in Materials and methods. Colonies surrounded by a clear zone were picked, and their 16S ribosomal DNA was sequenced to determine their taxonomic identities. The selected bacterial strains, including *Burkholderia gladioli*, *Burkholderia cepacia*, *Dyella japonica*, *Dyella yeojuensis*, *Pseudomonas aeruginosa*, and *Serratia marcescens*, were cultivated in the minimal medium, supplemented with 2% skim milk, at 28 °C for two days. The clarified broth after centrifugation was tested for peptidase activity by gliadin zymography, which enabled us to detect the number of the secreted peptidases and their relatively activities and molecular sizes. Among the bacteria, the *B. gladioli* strain was first selected for further study, because it secreted peptidase with stronger activity (Figure 1). The whole-genome sequence of the *B. gladioli* strain was then determined with pair-end sequencing using the Illumina Miseq system [15]. Accordingly, 7665 protein-coding sequences were predicted. Later, the secreted proteins of the *B. gladioli* strain were resolved on an SDS-PAGE gel, and the major bands were subjected to LC-tandem mass spectrometric analysis. A serine peptidase, termed Bga1903, was identified based on a search against those 7665 proteins. Bga1903 is a protein of 512 amino acids with a molecular mass of ~51.4 kDa. It consists of a sec-dependent signal peptide, a propeptide region and an enzymatic domain (Figure 2) according to the predictions by SignalP-5.0 and BLASTP analysis [16,17].

### 3.2. Heterologous Expression and Purification of Bga1903

The expression of Bga1903 in *E. coli* BL21(DE3) by using the authentic Bga1903 gene was poor, presumably due to the high GC content of the gene (70.4%). To overcome this hurdle, the unfavorable codons throughout the whole sequence of Bga1903 were modified according to the codon preference of *E. coli* translation system. This synthetic gene, termed Bga1903+, was inserted into plasmid pETDuet-1. In addition, a hexahistidine-coding sequence was added to the 3′ end of Bga1903+ to facilitate purification of the recombinant peptidase. After induction with 1 mM IPTG and overnight cultivation of the recombinant *E. coli* BL21(DE3) at 28 °C, a significant peptidase activity could be detected in the culture medium. The proteins in the medium were analyzed on an SDS-PAGE gel. It was obvious that several protein bands appeared when *E. coli* cells harbored pETDuet-Bga1903+ but not pETDuet-1 (Figure 3A). The clarified culture broth was concentrated, and the recombinant peptidase within was purified to homogeneity by immobilized metal affinity chromatography (IMAC) using a 5-mL HisTrap column (Figure 3B). It was also noted that some proteins with relatively large sizes were degraded by the peptidase activity during the concentration step. To define the boundary between the propeptide region and the enzymatic domain, the purified protein was subjected to Edman degradation analysis. The result showed an N-terminal LVP sequence, indicating that the mature Bga1903 starts from L162 (Figure 2). The purified mature Bga1903 was saved and used later for enzymatic characterization.

### 3.3. Gliadin Hydrolysis by Bga1903

To test whether the mature Bga1903 possesses an enzymatic activity for gliadin hydrolysis at acidic conditions, the protein was incubated with gliadin in 25 mM glycine-HCl buffer at pH 2.5, 3.0, or 3.5 at 37 °C for different time intervals. The gliadin content, in terms of size distribution, in each of the reaction solutions was analyzed by SDS-PAGE (Figure 4A). Hydrolysis of gliadin was obvious in the samples of 1.5- and 3-h incubation at pH 3.5 and of 3-h incubation at pH 3.0. Nonetheless, no gliadin degradation was observed in reactions at pH 2.5. Reduction in the molecular sizes of gliadin molecules does not necessarily represent the removal of the immunogenic peptides. Therefore, the hydrolyzed product of gliadin at various pH by the mature Bga1903 was further analyzed by the competitive ELISA method using R5 monoclonal antibody. The antibody primarily recognizes epitopes QQPFP, QQQFP, LQPFP and QLPFP, which are repetitively presented on the immunogenic peptides. The results indicate that the mature Bga1903 significantly removed the epitopes at pH 7.0 in a 1 h reaction at 37 °C (from 60 to 26 μg) (Figure 4B). Compared with the performance at pH 7.0, the residual gliadins decreased to 45.2 μg and 37.3 μg when the working pH was at pH 4.0 and 5.0, respectively. The results confirm that the mature Bga1903 can digest the immunogenic peptides of gliadin in acidic solution, although it works better at neutral pH.

### 3.4. Hydrolysis of Gluten-Derived Immunogenic Peptides by Bga1903

The presence of immunogenic peptides, such as the 33- and 26-mer peptides, is the culprit for celiac disease development. The known pro-epitopes include PFPQPQLPY, PYPQPQLPY, and PQPQLPYPQ of the 33-mer peptide and QQPFPQQPQ, QQPQQPYPQ, and QQPQQPFPQ of the 26-mer peptide. Thus, it was important to assess the enzymatic activity of the mature Bga1903 toward these pepsin-resistant peptides. Both 33- and 26-mer peptides were incubated with the mature Bga1903, followed by HPLC analysis. The HPLC chromatogram shows the potential of the mature Bga1903 to degrade both of the peptides (Figure 5A). The amino acid sequence of each proteolytic fragment was further analyzed by tandem mass spectrometry, and cleavage sites were assigned accordingly. The preferable scissile bonds were determined according to the count of every cleavage site relatively to the total count of all cleavages (Figure 5B). In general, the mature Bga1903 prefers to cut the peptide bonds at the carbonyl site of glutamine (P1 position). The peptide bond after leucine, tyrosine or phenylalanine was also degraded by the mature Bga1903 despite at a less frequency. According to the degradation patterns, the toxicity of the immunogenic pro-epitopes would be almost completely removed by the enzymatic activity of Bga1903.

### 3.5. Hydrolysis of BSA by Bga1903

To better understand the scissile bonds preferentially digested by Bga1903, BSA was treated with the purified Bga1903. The degraded fragments smaller than 10 kDa were analyzed by mass spectrometry. According to the amino acid sequence of each proteolytic fragment, every amino acid residue at the P1, P2 or P3 position was counted and normalized by its abundance in BSA as described in Materials and methods to obtain the amino acid preference at the P1, P2 and P3 positions (Figure 6). The probability of residues at the P1 position is in the order of K > F > L > Q, A, T > R, Y, H. For the P2 position, the preferential order is G > P > V > F, K, R > A, I, L. For P3, W and H are the two dominant residues, totally accounting for 50% probability. Obviously, Bga1903 is not a peptidase with a strict selection on the bonds to be cleaved. Nonetheless, the peptide bonds at the carbonyl site of positively charged or hydrophobic residues are generally favored. On the other hand, the peptide bonds after negatively charged residues are disfavored.

### 3.6. Activity toward Chromogenic Peptidyl Substrates

According to the preferential P1, P2 and P3 residues on BSA and immunogenic peptides, several dipeptidyl and tripeptidyl chromogenic substrates were synthesized (Table 1). Thus, the catalytic rate of Bga1903 could be measured according to the release rate of *p*-nitroaniline. Furthermore, the rates toward different substrates could be compared with each other. Among the group of Z-HHX-*p*NA, the reaction rate was in the order of Z-HHL-*p*NA > Z-HHK-*p*NA > Z-HHH-*p*NA, which is more or less consistent with the results obtained from BSA hydrolysis. Z-HHF-*p*NA was hardly hydrolyzed, presumably because it is less soluble than the other substrates. The substrate Z-HPQ-*p*NA, representing the scissile bond in gluten peptides, could be hydrolyzed at a relatively moderate rate. By contrast, Z-HAF-*p*NA, representing the peptide motif immediately in front of the mature Bga1903 domain (Figure 2), was hydrolyzed at a slower rate. This is in agreement with the concept that the propeptide must be cleaved only under certain condition so that the cell itself would not be harmed easily by its own peptidase. That Z-HYP-*p*NA and Z-QQP-*p*NA could not be hydrolyzed confirms the disfavor of proline at the P1 position. None of the dipeptidyl substrates was hydrolyzed, indicating that a minimum of three residues are required for substrate recognition by Bga1903. The catalytic constants of Bga1903 toward Z-HPQ-*p*NA were then determined through the measurement of the initial velocity under the substrate range of 0.05~1 mM at pH 7.0, 37 °C. According to the Lineweave-Burk plot, the values of *K*m and *k_cat_* were determined to be 0.44 ± 0.1 mM and 17.8 ± 0.4 s^−1^, respectively (Figure 7).

### 3.7. Application of Bga1903 in Beer Brewing

To be an effective oral therapeutic enzyme for celiac disease patients, the gluten-removing peptidase had better perform well in the stomach. Regarding this, the mature Bga1903 is hardly considered as an ideal candidate for this specific application. Nonetheless, the mature Bga1903 may still have an application potential for reducing the gluten content in processed foods. In this study, a commercial product of peptidases isolated from the culture medium of *A. niger* was added alone or with the mature Bga1903 into the wort immediately before the fermentation process. The adding amount of *A. niger* peptidases was 5 g and of Bga1903 was 0.012 g for 1 L wort. After completion of the fermentation, the gluten content in the clarified beer was measured by ELISA using R5 antibody. The gluten content in the control group, in which no peptidase was added, was 5230 μg/mL. However, it was reduced to 360 μg/mL if *A. niger* peptidases were included in the fermentation wort. It could be further reduced down to 1 μg/mL when the mature Bga1903 was additionally added. This gluten level is far below the 20-ppm threshold required for the gluten-free labeling. The final alcohol concentration in the beer was 7.5% in the control group. It increased slightly to 8.6% when peptidases were included during the fermentation, indicating that the added peptidases did not negatively affect the fermentation process.

## 4. Discussion

Given the prevalence of disorders related to gluten consumption, technologies such as the removal of gluten in processed foods and the oral therapy for celiac disease are desired. To eliminate celiac disease-toxic gluten peptides, a handful of peptidases from plants [18,19], fungi [20,21,22], and bacteria [21,23,24,25,26,27,28] have been studied. An excellent review has recently summarized the peptidases with potential in this regard [29]. Comparisons of the peptidases in the aspects of the original source, enzyme preparation, substrate preference, family classification and other characteristics are shown in Table 2.

EP-B2 is the cysteine endopeptidase B, isoform 2, from barley [18,30]. EP-B2 can hydrolyze the 33-mer peptide with a preference for peptide bonds after glutamine. Pro-EP-B2 was heterologously expressed in *E. coli* and a refolding process was adopted to obtain the mature EP-B2 in quantity. The fluid within the pitcher leaf of the carnivorous plant *Nepenthes* × *ventrata* is another source of plant peptidases to degrade gluten. Nepenthesin and neprosin extracted from the fluid can remove the 33-mer peptide at acidic conditions as low as pH 2.5 [19].

AN-PEP is an endopeptidase from *Aspergillus niger* [20,31,32]. It can remove the toxic gluten peptides with a preference for peptide bonds after proline at pH 4–5. In fact, AN-PEP has been marketed as a dietary supplement by DSM Nutritional Products (Heerlen, The Netherlands) under the name Tolerase^®^G [33,34]. Aspergillopepsin is another gluten-degrading peptidase from *A. niger*; nonetheless, it is not as substrate-specific and efficient as AN-PEP to remove the toxic gluten peptides [21,22]. DPP IV is a dipeptidyl exopeptidase isolated from *Aspergillus oryzae* [22,35]. Although it cleaves NH_2_-XP↓X- peptide bonds, it cannot hydrolyze the 33-mer peptide.

Bacteria offer a rich source of gluten-hydrolyzing peptidases. SC-PEP is an endopeptidase from *Sphingomonas capsulate* preferring the bonds after proline [36]. ALV003 is a formula containing EP-B2 and SC-PEP [18,23]. ALV003 can hydrolyze gluten efficiently in the stomach, as the two peptidases attack the peptide bonds with complementary preferences [37]. MX-PEP and FM-PEP are two other proline-preferring endopeptidases that were identified, respectively, from *Myxococcus xanthus* and *Flavobacterium meningosepticum* [23,38,39]. Subtilisin Carlsberg, produced by *Bacillus licheniformis*, is a well-studied peptidase [21,40]. With its broad substrate range, subtilisin Carlsberg also has an activity to hydrolyze gluten. However, this enzyme significantly underperforms at acidic conditions and is prone to autolysis. Through pharmaceutical modifications such as PEGylation and polylactic glycolic acid microencapsulation, the efficiency of subtilisin Carlsberg in removing the toxic gluten peptides at the acidic condition could be significantly improved [24,25]. Rothia subtilisin (Rmep) is a group of serine peptidases produced by *Rothia mucilaginosa*, a commensal bacterium naturally occurring in the human oral cavity [25,41,42,43]. Rothia subtilisin is capable of degrading the 33-mer peptide preferentially at the bonds after glutamine and tyrosine. Pseudolysin is a metalloendopeptidase produced by *Pseudomonas aeruginosa* [28]. It is capable of cleaving the 33-mer peptides preferentially at the bond -Q↓L-. Kuma030 is an iteratively engineered acid-tolerant S53 peptidase originally from *Alicyckobacillus sendaiensis* by Rosetta Molecular Modeling [26,44,45]. Kuma030 efficiently cleaves -PQ↓Q- and -PQ↓L- motifs of the toxic gluten peptides under a simulated gastric condition. Endopeptidase 40 (E40) is a serine protease originally produced by the soil actinomycete *Actinoallomurus* A8 [27]. E40 preferentially cleaves the peptide bond -Q↓L- of the 33-mer peptide with an optimum at pH 5.0.

In this study, we identified a serine peptidase, Bga1903, originally from the culture medium of a *B. gladioli* strain. The mature Bga1903 was secreted into the culture medium by the recombinant *E. coli*. The purified mature Bga1908 is capable of hydrolyzing the 33-mer and 26-mer peptides with a preference for the peptide bonds after glutamine; the bonds after leucine, tyrosine or phenylalanine are also susceptible, despite at rarer frequencies. In comparison with other reported gluten-hydrolyzing peptidases, Bga1903 shares the active site signatures of S8 subfamily with subtilisin Carlsberg and Rmep. As to the favorable scissile bonds on the toxic gluten peptides, Bga1903 exhibits the similar preference as EP-B2, subtilisin Carlsberg, Rmep, pseudolysin, Kuma030, and E40. As to the optimum pH, Bga1903 behaves better at pH 7.0, similarly to EP-B2, SC-PEP, MX-PEP, and pseudolysin.

Although Bga1903 preferred digesting the bonds -PQ↓L- and -PQ↓Q- on the toxic gluten peptides, it did not show a strict selection on the scissile bonds when BSA was hydrolyzed. Lysine, phenylalanine, leucine, and glutamine totally account for more than 50% of the probability at the P1 position. By contrast, glutamate and aspartate are disfavored. This preference suggests that the binding pocket S1 of Bga1903 is in a rather relaxed configuration and probably lined with a couple of negatively charged residues. As to the P2 position, glycine, proline and valine account for 40% probability, and arginine and lysine take another 20%. Therefore, the binding pocket S2 should be smaller than the S1. Moreover, the presumed negatively charged residues in the S1 also constitute the S2. This speculation, despite being reasonable, shall be validated by the crystal structure of Bga1903 in the near future.

As practical treatments for celiac disease, gluten-digesting enzymes have to overcome the harsh condition in human stomach with the presence of gastric acids and pepsin. In view of this, Bga1903 is not suitable for oral celiac disease therapy at its current version. Nonetheless, it may represent a good starting point for protein engineering or chemical modification to shift the activity range down to <pH 4.0. Besides being used as an oral therapy for celiac disease, gluten-degrading enzymes are useful to manufacture wheat flour-based gluten-free foods. With its excellent activity to remove the gluten-derived immunogenic peptides, Bga1903 may have a potential as a food additive to produce gluten-free foods as demonstrated in making gluten-free beer in this study. To further verify the effectiveness of Bga1903 in making gluten-free foods or beverages, Bga1903 should be prepared in large quantity in the future to address the issue of whether the application of Bga1903 alone is sufficient. The secretion of Bga1903 into the culture medium by the recombinant *E. coli* cells greatly simplifies the production and purification process of the peptidase. This advantage may facilitate the future application of Bga1903 in food and medical industries.

## Figures and Tables

**Figure 1 biomolecules-11-00451-f001:**
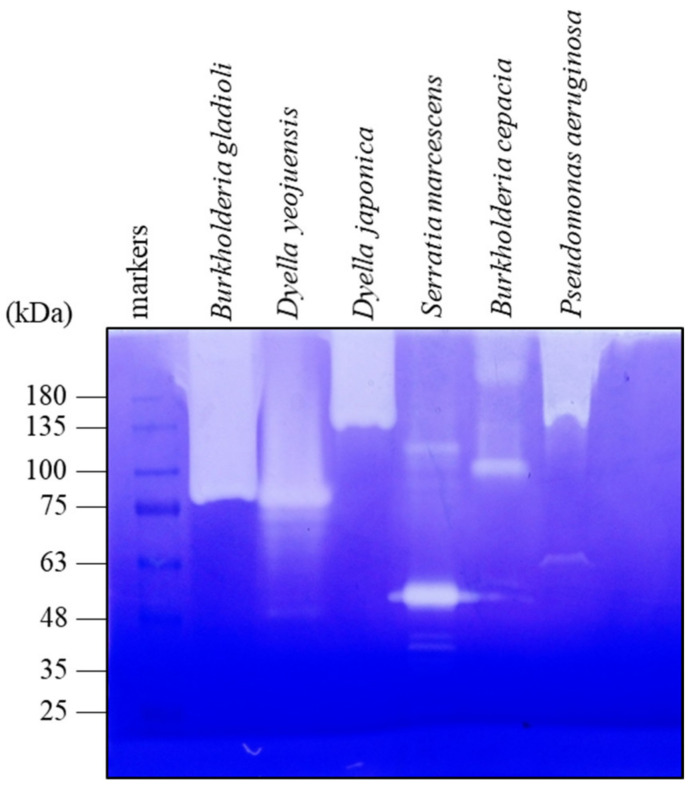
Gliadin zymogram of the culture broth of screened bacteria. The concentrated culture broth of the indicated bacterium was electrophoresed using a 2.2 mg/mL gliadin-containing polyacrylamide gel as described in Materials and methods. After electrophoresis, the gel was soaked twice in 100 mM Tris-HCl [pH 5.0] that contained 2.5% (*v*/*v*) Triton X-100 at 4 °C for 30 min, and subsequently incubated in 100 mM Tris-HCl [pH 5.0] that contained 1% (*v*/*v*) Triton X-100 at 37 °C for 1 h. The gel was then stained with *Coomassie* Brilliant *Blue* R-250 and distained following the regular SDS-PAGE procedure.

**Figure 2 biomolecules-11-00451-f002:**
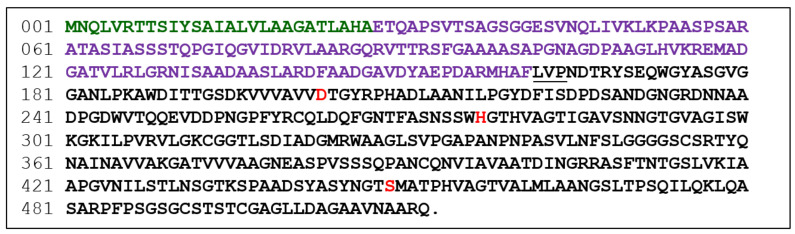
Domain organization of Bga1903. Amino acid residues constituting the sec-dependent signal peptide and the propeptide region are shown in green and purple, respectively, while those constituting the enzymatic domain are in black except the catalytic triad (D-H-S) that are shown in red. LVP, underlined, is the N-terminal sequence of the mature enzymatic domain according to the result of Edman degradation.

**Figure 3 biomolecules-11-00451-f003:**
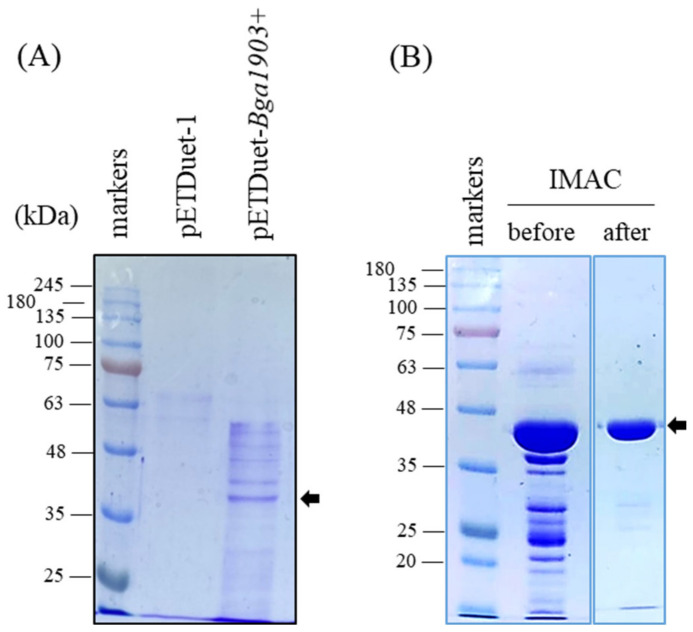
Expression and purification of Bga1903. (**A**) The recombinant peptidase was expressed in *E. coli* BL21(DE3) under the condition as described in Materials and methods. After 18 h culture at 28 °C, the clarified medium was concentrated 10-folds by ultrafiltration, and the proteins within were analyzed by SDS-PAGE. The *E. coli* cells that carried the plasmid pETDuet-1 served as the control. (**B**) The concentrated peptidase present in the medium was purified by IMAC as described in Materials and methods. The image data grouped in panel B were cropped from a single photo to have a clearer presentation. The arrows denote the mature Bga1903.

**Figure 4 biomolecules-11-00451-f004:**
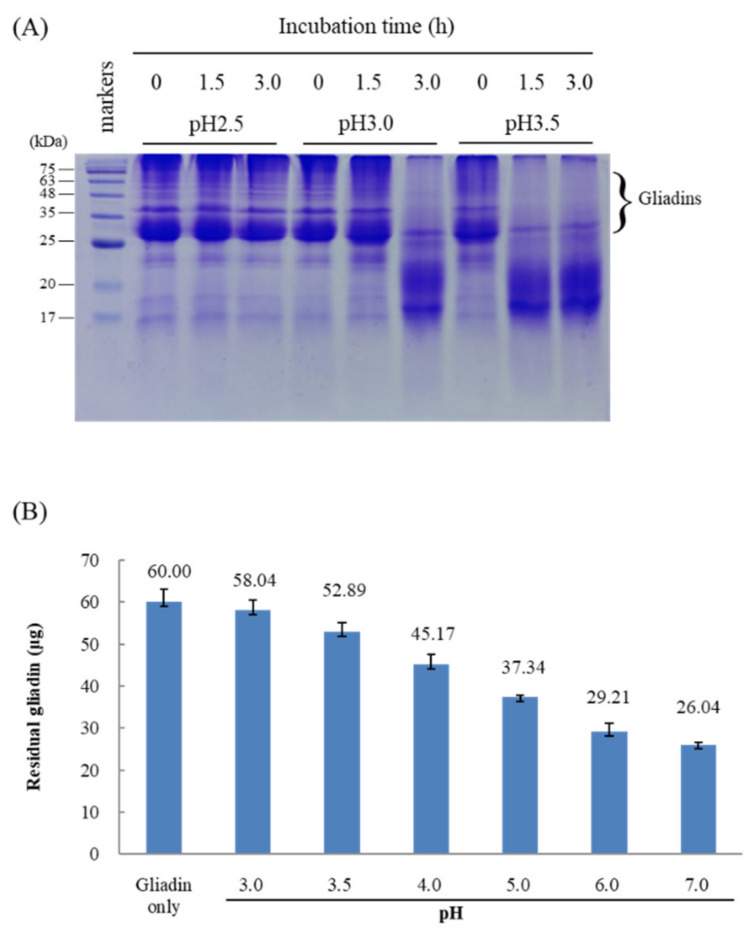
Hydrolysis of gliadin by Bga1903. (**A**) The mature Bga1903 (0.25 mg/mL) and gliadins (7.5 mg/mL) in glycine-HCl buffer at pH 2.5, 3.0, or 3.5 were incubated for up to 3 h. Hydrolysis of gliadins in each sample was analyzed by SDS-PAGE. (**B**) A 0.1 mL mixture that contained 60 μg gliadins, 1.2 μg mature Bga1903 and 50 mM citrate-phosphate buffer was incubated at the indicated pH, 37 °C, for 1 h. The residual gliadins in the final reaction solution were measured by a competitive ELISA using R5 monoclonal antibody, which specifically recognizes immunogenic epitopes on gliadins.

**Figure 5 biomolecules-11-00451-f005:**
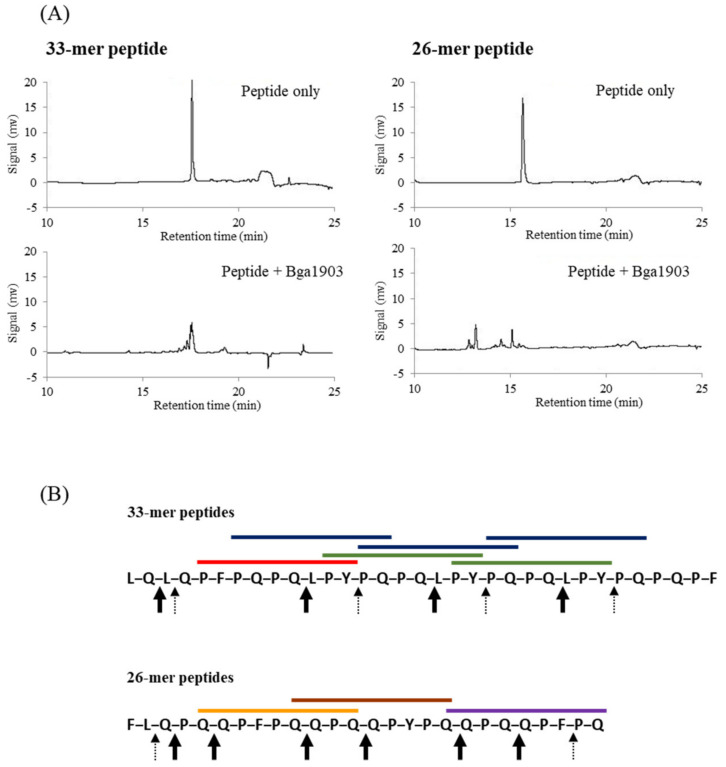
Hydrolysis of immunogenic peptides by Bga1903. (**A**) Degradation of the 33- and 26-mer peptides by the mature Bga1903 at pH 6.0 was analyzed by RF-HPLC according to the conditions described in Materials and methods. The sequences of degraded fragments were further determined by mass spectrometry. (**B**) The cleavage sites on the 33- and 26-mer peptides by Bga1903 are indicated by the black bold arrow, denoting major cleavages (>10% of total counts), and the dotted line arrow, denoting minor cleavages (<10% of total counts). The immunogenic pro-epitopes presented on the 33- and 26-mer peptides are shown with straight lines in various colors. For example, the three blue overlapped lines indicate the pro-epitope PQPQLPYPQ on the 33-mer peptide.

**Figure 6 biomolecules-11-00451-f006:**
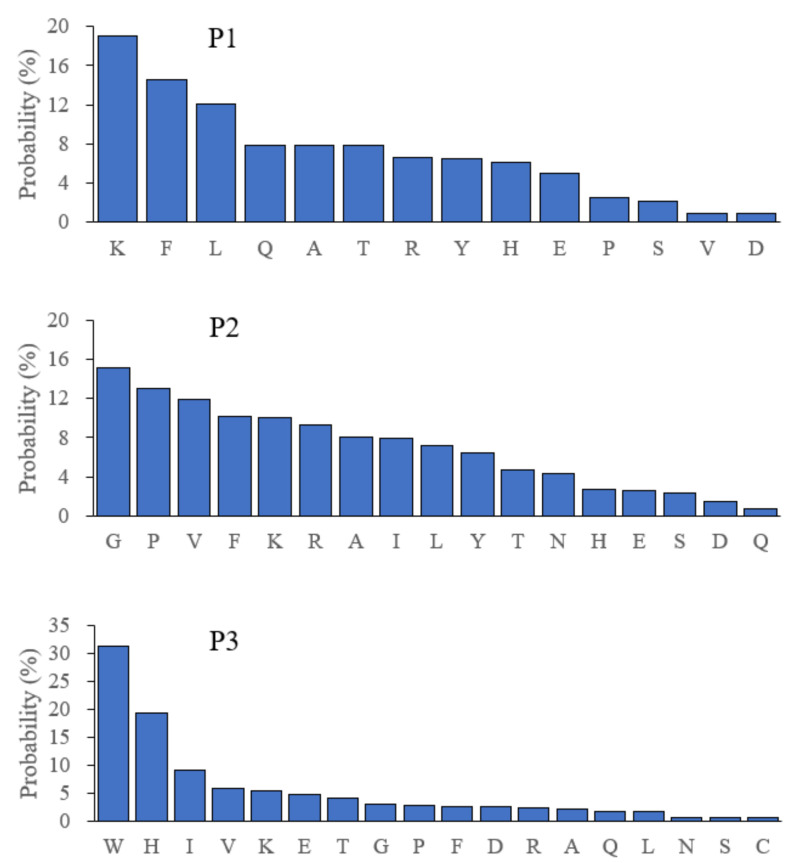
Preferential residues at the P1, P2 and P3 positions. BSA was digested by the mature Bga1903, and the cleavage sites were deduced from the proteolytic fragments determined by mass spectrometry as described in Materials and methods. The probability of given amino acid residues at the P1, P2, or P3 position was calculated as the description in Materails and methods. The preferential order for given residues at the P1, P2, or P3 position is listed from the left to the right.

**Figure 7 biomolecules-11-00451-f007:**
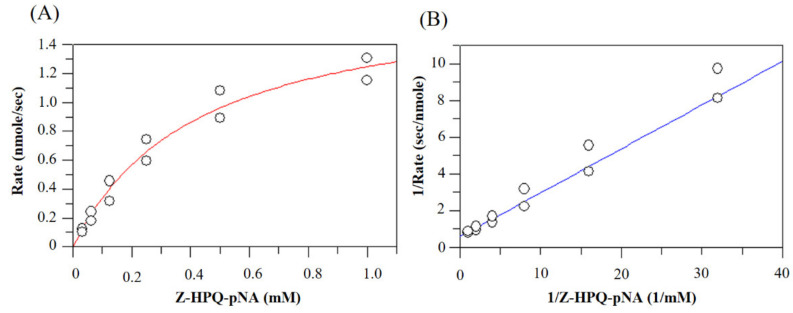
The substrate concentration dependence of catalytic rate of Bga1903. (**A**) The initial rate of releasing *p*-nitroaniline from peptidyl substrate Z-HPQ-pNA by the mature Bga1903 was measured at the reaction condition described in Materials and methods. (**B**) The double reciprocal plot from which the kinetic constants were calculated.

**Table 1 biomolecules-11-00451-t001:** The specific activity of the mature Bga1903 toward various chromogenic peptidyl substrates.

Chromogenic Substrate	Specific Activity (U/mg Mature Bga1903) ^a^
Z-HHL-pNA	1491
Z-HHK-pNA	483
Z-HPQ-pNA	280
Z-HHH-pNA	107
Z-HPF-pNA	72
Z-HAF-pNA	11
Z-HHF-pNA	3
Z-HYP-pNA	Not detectable
Z-QQP-pNA	Not detectable
Z-HH-pNA	Not detectable
Z-QP-pNA	Not detectable
Z-PP-pNA	Not detectable
Z-PY-pNA	Not detectable

^a^: The activity was measured at 37 °C, pH 7.0. One unit of activity was defined as the activity required to release 1 nmol *p*-nitroaniline per second.

**Table 2 biomolecules-11-00451-t002:** Reported peptidases with the proteolytic activities toward glutens.

Kingdom	Enzyme	Original Organism	Recombinant Host /Subcellular Location	Peptidase Family	Optimal pH	Preferable P1 (the 33-mer Peptide)	Ref
Plants	EP-B2	*Hordeum vulgare*	*E. coli* /inclusion body, refolding required	C1	7.0	Q	[18]
	nepenthesin, neprosin	*Nepenthes* × *ventrata*	pitcher fluid	Aspartic peptidase	2.5	P	[19]
Fungi	AN-PEP	*Aspergillus niger*	medium	S28	5.0	P	[20]
	Aspergillopepsin	*Aspergillus niger*	medium	A1	3.0	No activity to the 33-mer peptide	[22]
	DPP IV	*Aspergillus oryzae*	medium	S9	7.0	No activity to the 33-mer peptide	[22]
Bacteria	SC-PEP	*Sphigomonas capsulate*	*E. coli* /periplasm	S9	7.0	P	[23]
	MX-PEP	*Myxococcus xanthus*	*E. coli* /cytoplasm	S9	7.0	P	[23]
	FM-PEP	*Flavobacterium* *meningosepticum*	*E. coli* /cytoplasm	S9	8.0	P	[23]
	subtilisin Carlsberg	*Bacillus licheniformi*s	medium	S8	8.5	Q	[24]
	Rmep	*Rothia mucilaginosa*	cytoplasm	S8	9.0	Q	[25]
	pseudolysin	*Pseudomonas aeruginosa*	cytoplasm	M4	7.0	Q	[28]
	Kuma030	*Alicyckobacillus sendaiensis*	*E. coli* /cytoplasm	S53	4.0	Q	[26]
	E40	*Actinoallomurus* sp. A8	*Streptomyces lividans* /medium	S53	5.0	Q	[27]
	Bga1903	*Burkholderia gladioli*	*E. coli* /medium	S8	7.0	Q	This study

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
