# Peer review of "Efficient Hydrolysis of Gluten-Derived Celiac Disease-Triggering Immunogenic Peptides by a Bacterial Serine Protease from Burkholderia gladioli"

_biomolecules, 2021, doi:10.3390/biom11030451_

Round 1
Reviewer 1 Report
In this manuscript, Liu et al. identified Bga1903 is a serine endo-13 peptidase secreted by Burkholderia gladioli. The authors expessed Bga1903 in E. coli and generated secreted Bga1903, which was capable of hydrolyzing the gluten-derived toxic peptides.
This work provides compelling evidence that Bga1903 can decrease gluten content and could be used in food engineering.
Overall, the manuscript is well written and follows a logical flow. The conclusions are well supported by the findings and the discussion is appropriate.
As a result, I only have minor comments:
- Figure 1 shows Gliadin zymograms of Dyella, Serratia, and Pseudomonas, but these strains are not listed in the methods section. Please add the details of these cultures to the methods.
- It’s hard to read the inset in Figure 7- can this figure be made bigger and more legible
- Please define wort in the methods for those unfamiliar with the term
Author Response
1. Figure 1 shows Gliadin zymograms of Dyella, Serratia, and Pseudomonas, but these strains are not listed in the methods section. Please add the details of these cultures to the methods.
Screening and cultivation details of these strains were added at Page 2, lines 82-89. Bacterial strains mentioned in Figure 1 Gliadin zymogram are the ones with gliadin-hydrolyzing activity.
2. It’s hard to read the inset in Figure 7- can this figure be made bigger and more legible
Figure 7 (Pag. 11) was adjusted as two charts, making it more presentable.
3. Please define wort in the methods for those unfamiliar with the term.
The definition of wort is described at Page 4, lines 175-178, which was the extracted liquid from grain and malt. The more accurate recipe for the inoculation of yeast was also added at Page 4, lines 178-179.
Reviewer 2 Report
The Manuscript describes the identification, characterization and application of an endopeptidases produced by Burkholderia gladioli strain, Bga1903, for the hydrolysis of celiac disease immunogenic/toxic peptides. The Authors presented a detailed characterization of the enzyme and its hydrolytic activity. The limited hydrolytic activity showed by Bga1903 excluded its potential use for oral somministration, however, the Authors provided an alternative application in processed food such as the beer, to promote the relevance of the results.
I deem that the Manuscript boast good novelty and quality therefore it deserves pubblication after proper clarifications of few points itemized here in the following:
1) Figure 4B and relevant comments: In the Figure 4B the Authors reported the ug of gliadin removed and its % degree of removal. Does it mean for example that at pH 7 the original gliadin amount was 32.1 ug and it was totally removed (i.e. residual gliadin 0???)? I found this way to present the results a quite confusing and not really transparent, I would prefer to follow the trend of residual gliadin assayed after incubation at different pH. Please update the figure.
2) The Authors performed LC-MS/MS analysis to identify the AA sequence of the hydrolyzed peptides. Please provide more experimental details about how the sequence identification was carried out.
3) Pag. 12 lines 350-354: The Authors compare the residual gluten quantified by R5-ELISA after beer fermentation in presence of endopeptidases from A. niger and the combined presence of A. niger peptidases+Bga1903. The latter resulted in a clarified beer with only 1ug/mL residual gluten. Did the authors test the degradation efficiency of Bga1903 alone, meaning without A. niger peptidase? This experiment might be important to provide a full picture about the degradation efficiency of the Bga1903 during in a real food production process.
Author Response
1) Figure 4B and relevant comments: In the Figure 4B the Authors reported the ug of gliadin removed and its % degree of removal. Does it mean for example that at pH 7 the original gliadin amount was 32.1 ug and it was totally removed (i.e. residual gliadin 0???)? I found this way to present the results a quite confusing and not really transparent, I would prefer to follow the trend of residual gliadin assayed after incubation at different pH. Please update the figure.
Figure 4B (Page 8) was adjusted as the quantifications of residual gliadin after incubation with purified Bga1903 at different pH. The concentration without the interference of Bga1903 was 60 mg, and their respective residual gliadin was shown at the tops of the bars
2) The Authors performed LC-MS/MS analysis to identify the AA sequence of the hydrolyzed peptides. Please provide more experimental details about how the sequence identification was carried out.
The identification process of amino acid sequence was added at Page 4, lines 150-152 and Page 5, lines 167-169. The parameter for LC-tandem mass spectrometric analysis to determine of substrate preferences of Bga1903 was provided.
3) Pag. 12 lines 350-354: The Authors compare the residual gluten quantified by R5-ELISA after beer fermentation in presence of endopeptidases from A. niger and the combined presence of A. niger peptidases+Bga1903. The latter resulted in a clarified beer with only 1ug/mL residual gluten. Did the authors test the degradation efficiency of Bga1903 alone, meaning without A. niger peptidase? This experiment might be important to provide a full picture about the degradation efficiency of the Bga1903 during in a real food production process.
Deployment of Bga1903 alone would require large-scale quantity, for example 5 g, to achieve desired results, which was not feasible at the time when the experiment was performed. More tests will be conducted in the future to better represent the efficiency of Bga1903 in the processing of gluten-free foods and beverages.
Round 2
Reviewer 2 Report
The Authors revised the Manuscript according to the suggestions made in the first round.
As for the last point concerning the degradation efficiency of Bga1903 alone during beer fermentation, I understand the point raised by the Authors, however this remains an open issue of the investigation.
Therefore, for sake of clarity, I deem it is important to add a comment in the conclusions about this weak point, underlining the need for further investigations in a near future.
Author Response
We greatly appreciate your suggestion concerning the food-processing application of Bga1903 by itself.
The newly added statement is described on the line 9-12, the last paragraph of DISSCUSSION (page 14). We hope that this will avoid confusion for future readers.